# Clinical Evidence for Targeting NAD Therapeutically

**DOI:** 10.3390/ph13090247

**Published:** 2020-09-15

**Authors:** Dina Radenkovic, Eric Verdin

**Affiliations:** 1Health Longevity Performance Optimisation Institute, Cambridge CB22 5NE, UK; 2Fight Aging!, 4736 Onondaga Blvd, PMB 179, Syracuse, NY 13219, USA; reason@fightaging.org; 3Buck Institute for Research on Aging, Novato, CA 94945, USA; everdin@buckinstitute.org

**Keywords:** nicotinamide adenine dinucleotide, NAD, pharmacology

## Abstract

Nicotinamide adenine dinucleotide (NAD) pharmacology is a promising class of treatments for age-related conditions that are likely to have a favorable side effect profile for human use, given the widespread use of the NAD precursor vitamin B3 supplements. However, despite several decades of active investigation and numerous possible biochemical mechanisms of action suggested, only a small number of randomized and adequately powered clinical trials of NAD upregulation as a therapeutic strategy have taken place. We conducted a systematic review of the literature, following the PRISMA guidelines, in an attempt to determine whether or not the human clinical trials performed to date support the potential benefits of NAD supplementation in a range of skin, metabolic and age-related conditions. In addition, we sought medical indications that have yielded the most promising results in the limited studies to date. We conclude that promising, yet still speculative, results have been reported for the treatment of psoriasis and enhancement of skeletal muscle activity. However, further trials are required to determine the optimal method of raising NAD levels, identifying the target conditions, and comparisons to the present standard of care for these conditions. Lastly, pharmacological methods that increase NAD levels should also be directly compared to physiological means of raising NAD levels, such as exercise programs and dietary interventions that are tailored to older individuals, and which may be more effective.

## 1. Introduction

The cofactor nicotinamide adenine dinucleotide (NAD) is an important metabolic regulator of cellular redox reactions and a co-factor or a co-substrate for key enzymes essential for normal cellular function in different tissues. Known as NAD+ in its oxidized state and NADH in its reduced state, it was first described more than a century ago as a molecule in the electron transport chain in the metabolic reduction-oxidation reactions in mitochondria [1]. Poly(ADP-ribose) polymerases (PARPs), a group of enzymes that catalyze the transfer of ADP-ribose to target proteins, use NAD as a cofactor [2]. PARPs regulate many important cellular functions, including expression of transcription factors, gene expression and DNA repair. More recent interest in NAD emerged from research into the role of sirtuins, NAD-dependent deacylases, after the discovery that Sirtuin 2 is an NAD+-dependent histone deacetylase [3]. Sirtuins influence many important cellular processes, including inflammation, bioenergetics, circadian rhythm generation, and cell growth, all fundamental to cellular aging. These pathways place NAD at the center of cellular metabolism, mitochondrial function, and biological processes of aging.

Most human cells must rely on de novo creation of NAD from a variety of building blocks (Figure 1) [4]. NAD can be synthesized de novo from tryptophan via the kynurenine pathway or from nicotinic acid via the Preiss-Handler pathway [5]. However, the bulk of NAD synthesis in cells is generated via the NAD salvage pathways acting on the precursor molecule nicotinamide. Nicotinamide, the dominant NAD precursor, originates from the diet or can be produced by the activity of a variety of NAD hydrolases that include CD38/CD157, PARPs and Sirtuins [6].

NAD levels decline with increasing age, as the activities of both the salvage pathways and *de novo* synthesis are reduced, the result of altered levels of rate-limiting enzymes and precursors. There is also growing evidence that the activity of specific NAD hydrolases, particularly CD38, increases in specific tissues during aging [7]. Evidence from animal studies indicates that interventions that increase NAD levels produce numerous benefits on the overall cardiometabolic health and immune function [4]. A fall in NAD levels might be prevented with supplementation with NAD precursors, such as nicotinamide, nicotinic acid, nicotinamide mononucleotide (NMN), and nicotinamide riboside (NR). NMN and NR are believed to be orally bioavailable and feed into the NAD salvage pathway directly (NMN) or indirectly (NR via NMN) and thereby bypass a key rate-limiting step determined by the enzymatic activity of NAMPT which decreases with age. Both NMN and NR have received the most recent interest as promising future therapeutic strategies to raise NAD levels [8].

However, supplementation with NAD precursors is not the only way to increase NAD levels. Other possible approaches include strength training [9], upregulation of critical enzymes (e.g., rate-limiting NAMPT) involved in NAD salvage [10], and inhibiting NAD degradation (e.g., with bioavailable flavonoids to inhibit CD38) [11].

Evidence suggesting the benefits of NAD upregulation in humans has accumulated, but much of it involves the use of vitamin B3 and predates the most recent sirtuin-related expansion of knowledge of the role of NAD in cellular metabolism and aging. For example, daily niacin intake has been recommended for much of the past century. Furthermore, extensive literature already accompanies the use of high-dose niacin for dyslipidemias in the context of cardiovascular disease prevention [12,13]. However, the literature currently lacks a rigorous assessment of the potential role of NAD upregulation on healthspan using relevant surrogate biomarkers of aging.

A number of clinical trials have been conducted recently, with more underway, to rigorously assess NAD pharmacology in the context of aging and metabolic and age-related disease. These trials benefit from the more recent understanding of NAD and its relationships with important processes in cellular aging, and so NAD has been discussed as a therapeutic intervention for a range of age-related conditions, including immune decline (immunosenescence) and sterile chronic inflammation (inflammaging). De novo NAD synthesis is involved in the innate immune response of macrophages [14] and, indeed, the broader immune response, due to elevated cellular energy requirements during an acute immune response [15,16].

More recently, NAD upregulation has been proposed as a prevention or treatment of viral illnesses and vaccine boosters. These are of particular interest for treating COVID-19, a new disease caused by the novel SARS-CoV-2 coronavirus. The elderly are at increased risk of mortality and morbidity of the COVID-19 infection, and pharmacological enhancement of NAD levels, which might have a beneficial effect on the biological processes of aging, might be particularly useful in this context. Thus far, no evidence of benefit has been reported, and it remains unclear as to whether upregulating NAD is an effective therapy for immune decline. Animal studies have suggested a double-edged sword situation regarding NAD and immunity where raising NAD levels might lead to increased activity of immune cells in inflammatory conditions [15,17].

Considering all of the above, we have conducted a review of the literature in an attempt to determine whether or not the present human evidence for the potential benefits from NAD pharmacology supports an expansion of efforts to assess this approach to age-related conditions.

## 2. Methods

The review was conducted in accordance with the 2009 Preferred Reporting Items for Systematic Reviews and Meta-Analyses PRISMA Statement (Figure 2). MEDLINE (PubMed) and all databases across Web of Science were searched using search terms related to NAD supplementation in human studies and clinical outcomes. The search was limited to English language and adult human studies published up until 10 June 2020.

Using the title and abstract, we screened articles against inclusion and exclusion criteria. Subsequently, two independent authors (R, DR) independently reviewed full texts against eligibility criteria for final selection. Any disagreements between the two reviewers were resolved by discussion. Studies were excluded if they were non-English, were not conducted in human patients, or did not focus on increasing NAD levels. A record was retained if the full text was available, and the authors reported human trial data on the upregulation of NAD levels or targeting NAD biochemistry, such as via administration of NAD.

In addition, the existing literature and ClinicalTrials.gov were searched for pending or ongoing clinical trials of NAD+ pharmacology that have yet to publish results.

## 3. Results

### 3.1. Screening

In the initial screen, 289 records were identified. All were reviewed to ensure relevance and full-text availability. Of these, 36 records met the eligibility requirements. The characteristics of these studies are summarized in Table 1, in that the authors reported human trial data for interventions based on NAD pharmacology.

Additionally, 24 further clinical trials were identified as relevant to the assessment of NAD+ pharmacology, none of which is yet associated with published data. Most are recruiting or in progress, and are summarized in Table 2.

### 3.2. Summary of Studies

Of the 36 human trials with published results identified in our literature review, 18 reported on oral administration of NAD precursors, such as nicotinamide, NMN, or NR, while 8 employed oral administration of NAD. The remainder used an eclectic mix of exercise programs, antioxidants, forms of topical, intravenous, or intramuscular administration of NAD, as well as compounds targeting NQO1 activity. Of the 36 trials, 7 assessed only pharmacokinetics, safety, or biomarkers, and 17 reported beneficial outcomes. The remaining 12 reported no benefits to patients.

Our review reveals that the upregulation of NAD has been studied for a wide range of medical conditions, only a few of which are addressed by more than one study. These conditions include acute kidney injury, Alzheimer’s disease, chronic fatigue syndrome, dementia, hyperphosphatemia, hypertension, obesity, Parkinson’s disease, photoaging of skin, psoriasis, skin cancers, type 1 diabetes mellitus, type 2 diabetes mellitus, and schizophrenia.

NAD levels after intervention were measured in only 11 trials, in blood samples or tissues, via quantitative methods such as colorimetric analysis. In all cases, increased NAD was observed, but the size of the effect varied widely. These changes were almost incomparable due to the current lack of standardization, as variance may be due to differences in methodology of measurement, in interventions, or other factors.

Of the completed, published trials identified in this study, only two were adequately powered. Firstly, the European Nicotinamide Diabetes Intervention Trial that employed daily supplementation of nicotinamide. The primary outcome of progression to type 1 diabetes was negative [29]. Secondly, the open-label trial of oral and infused NADH in Parkinson’s disease patients by Birkmayer et al. showed a beneficial effect in 80% of the patients [23]. The effects on NAD in serum or tissues were assessed in neither of these studies. It should also be noted that the metabolism of intravenously infused NAD/NADH is currently unclear although it is likely that both metabolites are cleaved into nicotinamide and ADP-ribose by the liver, as has been demonstrated in mice [53]. It is also possible that CD38-expressing cells, such as leukocytes [54], act to degrade NAD/NADH in plasma.

#### 3.2.1. Neurological and Neurodegeneration Conditions

The earliest deliberate attempts at NAD pharmacology, as distinct from the extensive study of vitamin B3, involved the delivery of niacin or formulations of NAD in the treatment of schizophrenia, beginning in the 1960s [18,19,20]. This intervention was based on a variety of hypotheses linking NAD biochemistry to the neurobiological changes thought to be involved in schizophrenia. More modern hypotheses of NAD-related redox dysfunction in schizophrenia continue to be debated today; mitochondrial dysfunction and oxidative stress are thought to contribute to the pathogenesis of the condition [55]. Available reports refer to positive results in earlier studies, but the authors reported no benefits to patients resulting from their small clinical trials.

Beginning in the 1990s, NAD pharmacology was assessed as a basis for the treatment of Parkinson’s disease and Alzheimer’s disease [22,23,24,25], efforts that have since expanded to other forms of dementia [56]. To date, the results of clinical trials have been mixed for Parkinson’s disease and largely negative for Alzheimer’s disease. Further trials are in progress.

#### 3.2.2. Skin Conditions

NAD upregulation prevents actinic keratosis and improves some measures of photoaging [33,40]. While the mechanisms of action are not fully understood, NAD is a co-factor for the PARP enzymes that play a key role in DNA repair. Skin is exposed to UV damage, causing frequent DNA breaks. Improving PARP function, and thus improving DNA repair, might protect from precancerous skin lesions and other consequences of photoaging. This mechanism may also influence the skin pathology associated with dysregulated skin cell division in conditions, such as psoriasis [32].

#### 3.2.3. Metabolic Conditions

Only limited data are available for the use of NAD boosters in the treatment of metabolic conditions, such as obesity and metabolic syndrome. While some studies report improvement in lipid profile, exercise capacity, and muscle fiber composition despite a sedentary lifestyle [37,49], others show no benefit of supplementation in the prevention of type 1 diabetes [29], and no improvements in insulin resistance [44,47].

#### 3.2.4. Potential Side Effects

No study reported severe side effects, supporting the thesis that these interventions are likely to be relatively safe for human use. Multiple widely used over-the-counter supplements increase NAD levels, including niacin, NR, and NMN, and vitamin B3 supplementation have been in use for many years. Thus, it seems likely that side effects linked to interventions that target NAD metabolism more likely arise from impurities rather than the supplements themselves, since this industry generally operates without rigorous control of quality and standardization.

In short term experiments, high doses of nicotinamide cause hepatic toxicity [57]. High doses of niacin cause headaches, skin flushing, and dizziness [58]. These effects occur at doses much higher than those used in clinical trials and have so far not been reported in the limited number of studies of NAD pharmacology to date.

Long-term side-effects of NAD upregulation, harder to detect and quantify, may exist and sustained high doses of vitamin B3 compounds may have long-term side-effects. Excessive intake of niacin or nicotinamide may contribute to the development of Parkinson’s disease [57]. Nicotinamide may contribute to the development of diabetes, but animal data here contradict the human trial results showing no such effect [57]. Additionally, high doses of NR aggravate insulin resistance and produce white adipose tissue dysfunction in mice [59].

Another possible concern following chronic high-level administration of nicotinamide is the depletion of methyl groups. It has been reported that nicotinamide induces nicotinamide-N-methyltransferase which catalyzes the methylation of nicotinamide. This may cause secondary problems by consuming methyl groups that are required elsewhere to maintain cellular homeostasis [60].

NAD upregulation may also worsen the senescence-associated secretory phenotype (SASP) produced by senescent cells in old tissues. A proposed mechanism for this side-effect involves suppression of AMPK and p53, leading to stimulation of NF-κB via p38 MAPK and increased expression of inflammatory cytokines [61]. The burden of cellular senescence increases with age, and the SASP contributes to diverse pathologies of aging [62] An alternative hypothesis is that activation of a stronger SASP by NAD boosters may lead to enhanced clearance of senescent cells and an anti-aging effect, but supporting data, in either case, is presently very limited.

Supporting evidence for these potential long-term issues is sparse at best and entirely lacking in some cases. Given the fact that anti-aging interventions are likely to target comparatively healthy, comparatively younger individuals, quantification of potential long-term adverse effects will be critical to ensure that these interventions are safe. 

## 4. Discussion

NAD is required for critical cellular pathways involving NAD-consuming enzymes (sirtuins, PARPs and CD38) and NAD-utilizing but non consuming enzymes (using the NAD/NADH redox couple). These two classes of enzymes rely on effective NAD levels, giving this latter metabolite a broad influence over cellular biochemistry relevant to health, disease and aging. 

### 4.1. Gaps in the Knowledge of NAD

NAD levels decrease with age in tissues throughout the body. While numerous mechanisms have been proposed to explain this observation, there is still only an incomplete understanding of this phenomenon. Investigations of NAD biochemistry have largely focused on proximate causes, meaning changes in the expression of proteins that are rate-limiting in NAD synthesis and salvage pathways. For example, nicotinamide phosphoribosyltransferase (NAMPT) levels fall with age, and NAMPT is the rate-limiting enzyme for the pathway responsible for synthesizing NAD from nicotinamide [63].

Incomplete knowledge about the causative mechanisms of diminished NAD with aging leaves open the question of its connection to deeper causes of degenerative aging cataloged in the hallmarks of aging [64] and strategies for engineered negligible senescence (SENS) [65]. Further research is required to understand whether or not NAD pharmacology is in fact an efficient point of intervention, or whether deeper causes should be targeted instead.

Most researchers consider that for NAD or NAD precursors to exert beneficial effects, they must be taken up by cells to participate in pathways relevant to aging and age-related conditions. It remains to be robustly determined as to how oral or intravenous administration of the NAD produces benefits, as comparatively little work has taken place to characterize NAD+ transport into cells. While connexin 43 has been identified as a NAD+ transporter in a few cell lines and primary cell cultures [66], few groups have published on this topic. Evidence exists for extracellular NAD to influence processes relevant to osteoporosis and immunosenescence [67,68]. The literature points to participation in numerous other mechanisms of interest, as well. 

Notably, in this context, NMN enters cells via the Slc12a8 transporter [69]. Prior to this, it could only be hypothesized that such a mechanism existed, based on observing the results of NMN supplementation [70]. More remains to be discovered on how extracellular NAD and NAD precursors interact with cells, and such discoveries may resolve outstanding questions about the use of NAD or its precursors as an intervention.

### 4.2. Viruses and COVID-19

Cell and animal studies show that NAD is involved in regulating immune system activity [71,72]. Thus, NAD+ upregulation may ameliorate aspects of age-related immunosenescence and inflammaging [14], protect against viral disease and help the elderly to mount an immune response to vaccines. A trial of NR supplementation for COVID-19 in the elderly is recruiting [73].

However, although the animal evidence is promising, more human clinical studies are required. We do not know which forms of NAD pharmacology work to address immune function, nor is the dose range established. Further, it is unknown as to whether it would be beneficial to upregulate NAD via supplements as an immune booster for the entire elderly population, or only as a specific treatment for certain medical conditions.

## 5. Conclusions

Based on the human trials conducted to date, NAD pharmacology is a promising treatment strategy that is likely to be safe for human use. However, despite several decades of active investigation, there is still only suggestive evidence, in the form of a few successful and sufficiently powered clinical trials, for NAD upregulation to be effective for any of the many potential indications where it may benefit patients.

Further trials are justified: about half of the published trials identified in our review reported patient benefits. Most of the indications tested require further studies for replication because only a single trial has taken place.

In general, it is the case that more and larger studies are required to produce robust data in support of NAD pharmacology. This includes in particular studies in which different forms of NAD upregulation are compared consistently with one another. For example, exercise programs tailored to older individuals may be more effective than all of the existing approaches to NAD pharmacology. Whether or not this is the case is one of the more important questions for the research community to answer.

## Figures and Tables

**Figure 1 pharmaceuticals-13-00247-f001:**
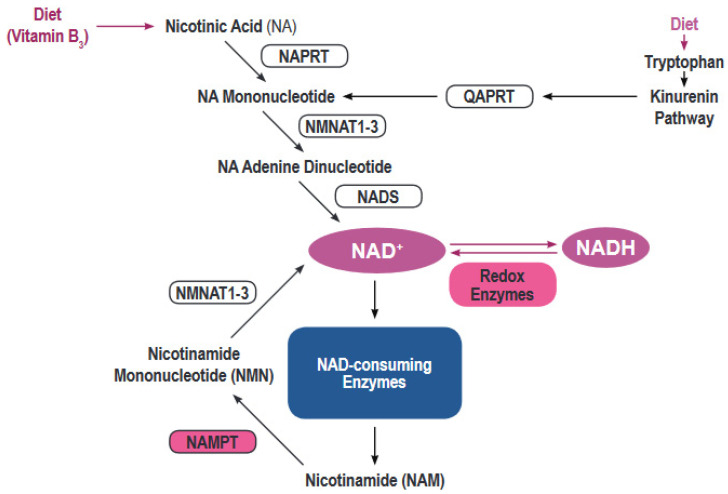
Regulation of the nicotinamide adenine dinucleotide (NAD) metabolism. The figure shows the main synthesis pathways of NAD including the de novo synthesis via the kynurenine pathway or from nicotinic acid and the salvage pathways.

**Figure 2 pharmaceuticals-13-00247-f002:**
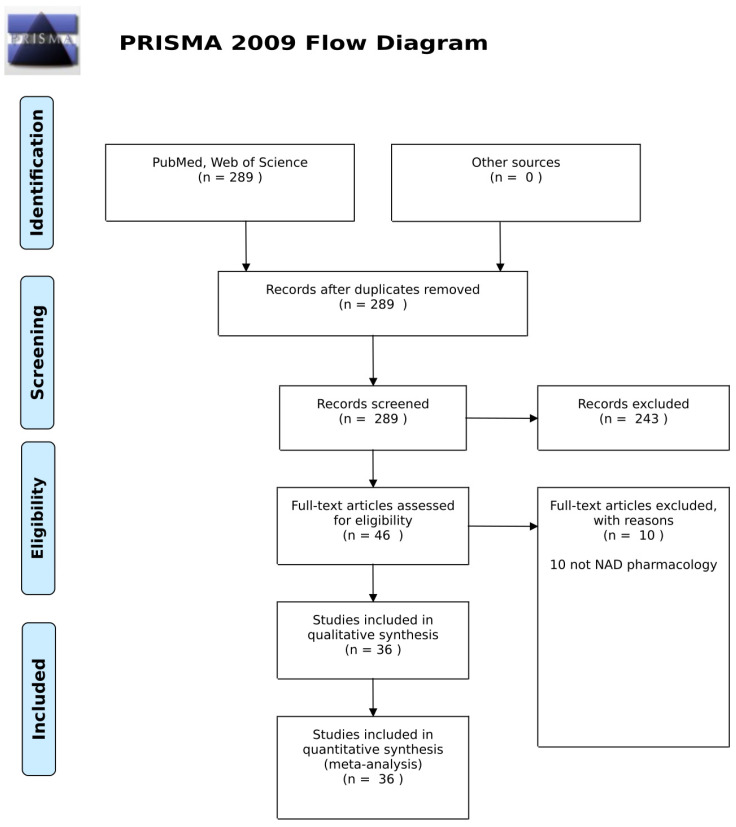
PRISMA 2009 flow diagram.

**Table 1 pharmaceuticals-13-00247-t001:** Studies of NAD pharmacology and related interventions.

Study	Intervention	N
Controlled evaluation of nicotinamide adenine dinucleotide in the treatment of chronic Schizophrenic patients, Kline et al., 1967 [18] Oral NAD 1 or 2 g daily. No systemic improvement in either treatment group.	NAD	14
Diphosphopyridine nucleotide in the treatment of schizophrenia, Kline et al., 1967 [19] Oral NAD 1 g or 2 g daily. No positive results were seen.	NAD	20
The behavioral effects of nicotinamide adenine dinucleotide in chronic schizophrenia, Meltzer et al., 1969 [20] Oral 2 g NAD daily for 21 days. NAD: After a single dose in volunteer blood samples, not in the study population. No gross clinical improvement was noted.	NAD	10
Nicotinic acid in the treatment of schizophrenia, Ban, 1975 [21] Oral 3000 mg nicotinic acid daily. Inferior to placebo.	Nicotinic acid	86
The coenzyme nicotinamide adenine dinucleotide (NADH) improves the disability of parkinsonian patients, Birkmayer et al., 1989 [22] Intravenous infusion of 25 mg NADH daily for 10–14 days. 21 patients (61.7%) showed a very good (better than 30%) improvement of disability; 13 patients (38.3%), a moderate (up to 30%) improvement.	NADH	34
Nicotinamide adenine dinucleotide (NADH)--A new therapeutic approach to Parkinson’s disease. Comparison of oral and parenteral application, Birkmayer et al., 1993 [23] Intravenous infusion and oral NADH, 25–50 mg/day. 19.3% of the patients showed a very good (30–50%) improvement of disability; 58.8% a moderate (10–30%) improvement; 21.8% did not respond to NADH.	NADH	885
Treatment of Parkinson’s disease with NADH, Dizdar et al., 1994 [24] Intramuscular 25 mg NADH on days 1–4, 14, 28. No statistically significant benefit.	NADH	9
Coenzyme nicotinamide adenine dinucleotide: New therapeutic approach for improving dementia of the Alzheimer type, Birkmayer, 1996 [25] Oral 10 mg/day NADH for 8–12 weeks. Improvement in mini-mental state examination and global deterioration scale. This open-label trial represents a pilot study from which no definitive conclusion can be drawn.	NADH	17
Nicotinic acid supplementation: Effects on niacin status, cytogenetic damage, and poly(ADP-ribosylation) in lymphocytes of smokers, Hageman et al., 1998 [26] Oral 0–100 mg/day nicotinic acid. NAD+: Small increase in PBMC. No evidence was found for a decrease in cigarette smoke-induced cytogenetic damage.	Nicotinic acid	21
Therapeutic effects of oral NADH on the symptoms of patients with chronic fatigue syndrome, Forsyth et al, 1999 [27] Oral 10 mg NADH daily. A significant 31% favorable response to NADH versus 8% to placebo.	NADH	26
No evidence for cognitive improvement from oral nicotinamide adenine dinucleotide (NADH) in dementia, Rainer et al., 2000 [28] Oral 10 mg NADH daily for 12 weeks. No evidence for any cognitive effect as defined by established psychometric tests.	NADH	25
European Nicotinamide Diabetes Intervention Trial (ENDIT): A randomized controlled trial of intervention before the onset of type 1 diabetes, Gale et al., 2004 [29] Oral 1.2 g/m^2^ nicotinamide daily up to a maximum of 3 g/day for 5 years in two divided doses. No difference in the development of diabetes between the treatment groups.	Nicotinamide	552
Nicotinamide effects oxidative burst activity of neutrophils in patients with poorly controlled type 2 diabetes mellitus, Osar et al., 2004 [30] Oral 50 mg/kg nicotinamide daily for 1 month. Oxidative burst activity was greater in patients receiving nicotinamide.	Nicotinamide	30
Nicotinamide suppresses hyperphosphatemia in hemodialysis patients, Takahashi et al., 2004 [31] Oral 500 mg nicotinamide daily for 12 weeks increasing by 250 mg every 2 weeks. NAD: ~42% increase in blood levels. Increased serum HDL cholesterol, decreased LDL cholesterol and phosphate.	Nicotinamide	65
In search for new antipsoriatic agents: NAD+ topical composition, Wozniacka et al., 2006 [32] Topical 1% or 0.3% NAD+ daily for 4 weeks. Topical NAD+ has an antipsoriatic potential, similar to that of anthralin.	NAD+	37
A topical lipophilic niacin derivative increases NAD, epidermal differentiation and barrier function in photodamaged skin, Jacobson et al., 2007 [33] Topical myristyl nicotinate 1–5% daily for 8–18 weeks. NAD: ~125% increase in the skin. MN treatment of photodamaged facial skin increased stratum corneum thickness by 70% and epidermal thickness by approximately 20%.	Myristyl nicotinate	16 and 60
Skeletal muscle NAMPT is induced by exercise in humans, Costford et al., 2010 [34] Exercise: alternating day progressive 30–60-min interval protocol and 50-min aerobic protocol for 3 weeks. NAMPT protein increased in sedentary nonobese subjects.	Aerobic exercise	13
The effect of antioxidant supplementation on fatigue during exercise: Potential role for NAD + (H), Mach et al., 2010 [35] Oral 0.36 mg pycnogenol 3 h prior to exercise. AD and NADH increased in muscle and serum. An increase of 17% in physical work capacity until fatigue.	Pycnogenol	13
Epigenetic and neurological effects and safety of high-dose nicotinamide in patients with Friedreich’s ataxia: An exploratory, open-label, dose-escalation study, Libri et al., 2014 [36] Oral 2–8 g nicotinamide for 8 weeks. A sustained improvement in frataxin concentrations. Clinical measures showed no significant changes.	Nicotinamide	40
Evidence for a direct effect of the NAD+ precursor Acipimox on muscle mitochondrial function in humans, van de Weijer et al., 2014 [37] Oral 250 mg Acipimox for 2 weeks. A rebound rise in plasma NEFA, negatively impacted insulin sensitivity. Skeletal muscle mitochondrial oxidative capacity and ATP production improved.	Acipimox	21
Effect of coenzyme Q 10 plus nicotinamide adenine dinucleotide supplementation on maximum heart rate after exercise testing in chronic fatigue syndrome, A randomized, controlled, double-blind trial, Castro-Marrero et al., 2015 [38] Oral 50 mg of CoQ10 and 5 mg of NADH twice daily for 8 weeks. NADH: ~252% increase in PBMC. A significant reduction in max HR during a cycle ergometer test at week 8 versus baseline.	CoQ10 / NADH	80
An open-label, non-randomized study of the pharmacokinetics of the nutritional supplement nicotinamide riboside (NR) and its effects on blood NAD+ levels in healthy volunteers, Airhart et al., 2017 [39] Oral 250–1000 mg NR twice daily for 8 days. NAD+: ~100% increase in whole blood. NR was safe and well-tolerated.	NR	8
Prevention of non-melanoma skin cancers with nicotinamide in transplant recipients: a case-control study, Drago et al., 2017 [40] Oral 500 mg nicotinamide daily. Actinic keratoses (AKs) significantly decreased in size in 18/19 patients (88%). In controls, 91% showed an increase in size or number of AKs.	Nicotinamide	38
Pharmacokinetics and tolerability of MB12066, a beta-lapachone derivative targeting NAD(P)H: quinone oxidoreductase 1: two independent, double-blind, placebo-controlled, combined single and multiple ascending doses first-in-human clinical trials, Kim et al., 2017 [41] Oral MB12066 up to a 400-mg single dose or 200 mg daily for 7 days. MB12066 was safe and well-tolerated.	MB12066	56 and 20
Pharmacokinetic and safety evaluation of MB12066, an NQO1 substrate, Lee et al., 2017 [42] Oral 100 mg MB12066 twice daily. MB12066 was safe and well-tolerated.	MB12066	8 and 3
Phase II clinical trial of nicotinamide for the treatment of mild to moderate Alzheimer’s disease, Phelan et al., 2017 [43] Oral 1500 mg nicotinamide twice daily for 24 weeks. There were no significant effects of nicotinamide on the primary or secondary endpoints.	Nicotinamide	31
A randomized placebo-controlled clinical trial of nicotinamide riboside in obese men: safety, insulin-sensitivity, and lipid-mobilizing effects, Dollerup et al., 2018 [44] Oral 1000 mg NR twice daily for 12 weeks. No improvement in insulin sensitivity, endogenous glucose production, glucose disposal and oxidation, resting energy expenditure, lipolysis, oxidation of lipids, or body composition.	NR	40
Chronic nicotinamide riboside supplementation is well-tolerated and elevates NAD + in healthy middle-aged and older adults, Martens et al., 2018 [45] Oral NR 500 mg twice daily. NAD+: ~60% increase in PBMC. NR is well tolerated and effectively stimulates NAD + metabolism.	NR	30
De novo NAD+ biosynthetic impairment in acute kidney injury in humans, Mehr et al., 2018 [46] Oral 1 or 3 g nicotinamide daily for 3 days. Nicotinamide administration was not associated with increased adverse events compared to placebo.	Nicotinamide	55
Effects of nicotinamide riboside on endocrine pancreatic function and incretin hormones in nondiabetic men with obesity, Dollerup et al., 2019 [47] Oral 1000 mg NR twice daily for 12 weeks. No effect on fasting or post-glucose challenge concentrations of glucose, insulin, C-peptide, glucagon, GLP-1, or GIP. β-cell function did not respond to the intervention. No change in circulating adipsin or bile acids after NR supplementation.	NR	40
Acute nicotinamide riboside supplementation improves redox homeostasis and exercise performance in old individuals: a double-blind cross-over study, Dolopikou et al., 2019 [48] Oral 500 mg NR, one dose. NR supplementation significantly increased NADH and NADPH, improved isometric peak torque by 8%, and fatigue index by 15%.	NR	24
Nicotinamide riboside augments the aged human skeletal muscle NAD+ metabolome and induces transcriptomic and anti-inflammatory signatures, Elhassan et al., 2019 [49] Oral 1 g NR daily for 21 days. NR elevated the muscle NAD+ metabolome; downregulated energy metabolism and mitochondria pathways without altering mitochondrial bioenergetics; depressed levels of circulating inflammatory cytokines.	NR	12
Effect of oral administration of nicotinamide mononucleotide on clinical parameters and nicotinamide metabolite levels in healthy Japanese men, Irie et al., 2019 [50] Oral 100–500 mg NMN, single dose. Safe and effectively metabolized in healthy men without causing any significant deleterious effects.	NMN	10
Efficacy and tolerability of EH301 for amyotrophic lateral sclerosis: A randomized, double-blind, placebo-controlled human pilot study, de la Rubia et al., 2019 [51] 600 mg EH301 twice daily for 4 months. EH301 was shown to significantly slow the progression of ALS relative to placebo.	EH301	32
Resistance training increases muscle NAD+ and NADH concentrations as well as NAMPT protein levels and global sirtuin activity in middle-aged, overweight, untrained individuals, Lamb et al., 2020 [9] 10 weeks of full-body resistance training. Muscle NAD+, NADH, and global SIRT activity are positively affected by resistance training in middle-aged, untrained individuals.	Resistance training	16
Niacin cures systemic NAD + deficiency and improves muscle performance in adult-onset mitochondrial myopathy, Pirinen et al., 2020 [52] Oral niacin, 750–1000 mg daily, 4 months for controls, 10 months for patients. Blood NAD+ increased eightfold in controls and patients. Muscle NAD+ of patients reached that of controls. Muscle strength and biogenesis increased in controls and patients. Muscle metabolome of patients shifted towards controls. Liver fat decreased 50% in patients.	Niacin	15

**Table 2 pharmaceuticals-13-00247-t002:** Selected clinical trials yet to publish results.

Trial	Intervention	N
Effect of “nicotinamide mononucleotide” (NMN) on cardiometabolic function (NMN) NCT03151239	NMN	25
Effect of long-term oral administration of nicotinamide mononucleotide (NMN) on human health UMIN000025739	NMN	20
Assessment of the safety of long-term nicotinamide mononucleotide (NMN). UMIN000030609	NMN	30
Evaluate the efficacy and safety of uthever NMN (nicotinamide mononucleotide, a form of vitamin B3) (NMN) NCT04228640	NMN	66
Effects of vitamin B3 derivative nicotinamide Riboside (NR) in bone, skeletal muscle and metabolic functions in aging NCT03818802	NR	48
Nicotinamide riboside for treating elevated systolic blood pressure and arterial stiffness in middle-aged and older adults NCT03821623	NR	118
Nicotinamide riboside in hospitalized patients NCT04110028	NR	84
The effects of nicotinamide adenine dinucleotide (NAD) on brain function and cognition (NAD) NCT02942888	NR	46
Crossover trial for nicotinamide riboside in subjective cognitive decline and mild cognitive impairment NCT04078178	NR	40
Evaluation of nicotinamide riboside in the prevention of small fiber axon degeneration and promotion of nerve regeneration NCT03912220	NR	40
NAD therapy for improving memory and brain blood flow in older adults with mild cognitive impairment NCT03482167	NR	58
A randomized controlled trial of nicotinamide supplementation in early Parkinson’s disease (NOPARK) NCT03568968	NR	400
NAD-supplementation in drug-naive Parkinson’s disease (NAD-PARK) NCT03816020	NR	30
Effects of vitamin B3 in patients with ataxia-telangiectasia NCT03962114	NR	24
Nicotinamide riboside in systolic heart failure NCT03423342	NR	30
Study to evaluate the effect of nicotinamide riboside on immunity NCT02812238	NR	38
Nicotinamide riboside and mitochondrial biogenesis NCT03432871	NR	15
Effects of nicotinamide riboside on the clinical outcome of Covid-19 in the elderly (NR-COVID19) NCT04407390	NR	100
A study to evaluate the safety and health benefits of Basis™ among elderly subjects. (15BSHE) NCT02678611	Basis (NR, pterostilbene)	120
A study by ChromaDex to assess the effects of TRU NIAGEN on cognitive function, mood and sleep in older adults NCT03562468	TRU NIAGEN (NR)	40
NAD Supplementation Study (NADS) NCT03310034	Nicotinic acid, nicotinamide, tryptophan	14
Nicotinamide as an early Alzheimer’s disease treatment (NEAT) NCT03061474	Niacin, nicotinamide	48
Study of the efficacy and safety of nicotinamide in patients with Friedreich ataxia (NICOFA) NCT03761511	Nicotinamide	225
Niacin for Parkinson’s disease (NAPS) NCT03808961	Niacin	100

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
