# Peer review of "Clinical Evidence for Targeting NAD Therapeutically"

_pharmaceuticals, 2020, doi:10.3390/ph13090247_

Round 1

Reviewer 1 Report

NAD has recently come into focus as a therapeutucally relevant molecule with potential anti-aging activity and other beneficial properties. Therefore, clinical studies that verify the therapeutic potential of NAD are of high interest. Adenkovic et al. reviewed the completed clinical trials investigating the potential therapeutic effects of increasing NAD levels as well as selected ongoing trials on the same subject. The review has been systematically conducted using a clear and well-reported  methodology. The results are organized in a logical way, clearly presented and adequatly discussed. The manuscript is creally written and easy to follow. I think the review will be interesting  to researchers working in many different NAD-related fields. I recommend publication after minor revision of Fig. 1.

Minor comment: Figure 1 depicts NAD structure from wikipedia. Perhaps it would be useful to replace this with an originally drawn structure and add other NAD-related compounds that are often mentioned in the text (NADH, NR etc.)

Author Response

Thank you very much for your suggestion. We have now drawn a new Figure 1 illustrating NAD as well as the main NAD-synthesis pathways and related metabolites. 

Reviewer 2 Report

In this review, the Authors summarized the clinical trials conducted with NAD, NAD precursors, or other means to potentiate NAD levels. I believe this review is clearly written and focused: the Authors discuss what can be drawn from the completed trials and, especially, they highlight how much still has to be done and clarified regarding the beneficial effects of enhancing NAD levels to counteract aging and related disorders.

I recommend the publication of this review in the present form, after addressing these points:

Lines 140-141: Authors should, together with liver, also mention the possibility that NAD is degraded in the plasma by CD38-expressing cells (indeed, some types of leukocytes express high level of CD38, and CD38 is also expressed by red blood cells, ref. PMID: 8250903).

Line 141: ADP-ribose is misspelled.

Line 222: Authors should mention a study reporting that connexin 43 hemichannels can transport NAD (PMID: 11099492). This observation has been subsequently confirmed by independent groups. Thus, stating that “it is unclear how intact NAD can be taken up by cells” does not acknowledge work done by others.

Author Response

Thank you very much for your comments and your help in improving this manuscript. 

Comment 1: Lines 140-141: Authors should, together with liver, also mention the possibility that NAD is degraded in the plasma by CD38-expressing cells (indeed, some types of leukocytes express high level of CD38, and CD38 is also expressed by red blood cells, ref. PMID: 8250903).

Thank you for your comment. We have now added the following to this paragraph: "It is also possible that CD38-expressing cells, such as leukocytes, act to degrade NAD/NADH in plasma." with a reference to https://doi.org/10.1006/bbrc.1993.2416 as suggested. 

-----------------------

Comment 2: Line 141: ADP-ribose is misspelled. Thank you for your comment. We have now corrected the spelling: ADP-ribosed has been switched to ADP-ribose.

-----------------------

Comment 3: Line 222: Authors should mention a study reporting that connexin 43 hemichannels can transport NAD (PMID: 11099492). This observation has been subsequently confirmed by independent groups. Thus, stating that “it is unclear how intact NAD can be taken up by cells” does not acknowledge work done by others.

Thank you for your comment. We have now included that work and elaborated on NAD uptake by the cells. The sentence  "it is unclear how intact NAD can be taken up by cells" has been changed to "comparatively little work has taken place to characterize NAD+ transport into cells. While connexin 43 has been identified as a NAD+ transporter in a few cell lines and primary cell cultures, few groups have published on this topic." Reference added to https://doi.org/10.1096/fj.00-0566fje.